# Evaluation of Vertical Ground Reaction Forces Pattern Visualization in Neurodegenerative Diseases Identification Using Deep Learning and Recurrence Plot Image Feature Extraction

**DOI:** 10.3390/s20143857

**Published:** 2020-07-10

**Authors:** Che-Wei Lin, Tzu-Chien Wen, Febryan Setiawan

**Affiliations:** 1Department of Biomedical Engineering, College of Engineering, National Cheng Kung University, Tainan 701, Taiwan; lkkcharlie.wtmh@gmail.com (T.-C.W.); febryans2802.wtmh@gmail.com (F.S.); 2Medical Device Innovation Center, National Cheng Kung University, Tainan 701, Taiwan

**Keywords:** gait analysis, pattern visualization, neurodegenerative diseases, deep learning, feature extraction, recurrence plot, vertical ground reaction force (vGRF) data

## Abstract

To diagnose neurodegenerative diseases (NDDs), physicians have been clinically evaluating symptoms. However, these symptoms are not very dependable—particularly in the early stages of the diseases. This study has therefore proposed a novel classification algorithm that uses a deep learning approach to classify NDDs based on the recurrence plot of gait vertical ground reaction force (vGRF) data. The irregular gait patterns of NDDs exhibited by vGRF data can indicate different variations of force patterns compared with healthy controls (HC). The classification algorithm in this study comprises three processes: a preprocessing, feature transformation and classification. In the preprocessing process, the 5-min vGRF data divided into 10-s successive time windows. In the feature transformation process, the time-domain vGRF data are modified into an image using a recurrence plot. The total recurrence plots are 1312 plots for HC (16 subjects), 1066 plots for ALS (13 patients), 1230 plots for PD (15 patients) and 1640 plots for HD (20 subjects). The principal component analysis (PCA) is used in this stage for feature enhancement. Lastly, the convolutional neural network (CNN), as a deep learning classifier, is employed in the classification process and evaluated using the leave-one-out cross-validation (LOOCV). Gait data from HC subjects and patients with amyotrophic lateral sclerosis (ALS), Huntington’s disease (HD) and Parkinson’s disease (PD) obtained from the PhysioNet Gait Dynamics in Neurodegenerative disease were used to validate the proposed algorithm. The experimental results included two-class and multiclass classifications. In the two-class classification, the results included classification of the NDD and the HC groups and classification among the NDDs. The classification accuracy for (HC vs. ALS), (HC vs. HD), (HC vs. PD), (ALS vs. PD), (ALS vs. HD), (PD vs. HD) and (NDDs vs. HC) were 100%, 98.41%, 100%, 95.95%, 100%, 97.25% and 98.91%, respectively. In the multiclass classification, a four-class gait classification among HC, ALS, PD and HD was conducted and the classification accuracy of HC, ALS, PD and HD were 98.99%, 98.32%, 97.41% and 96.74%, respectively. The proposed method can achieve high accuracy compare to the existing results, but with shorter length of input signal (Input of existing literature using the same database is 5-min gait signal, but the proposed method only needs 10-s gait signal).

## 1. Introduction

Neurodegenerative diseases (NDDs), such as amyotrophic lateral sclerosis (ALS), Huntington’s disease (HD) and Parkinson’s disease (PD), are caused by malfunctioning neurons in different regions of the nervous system [1]. PD, the second most common NDD, is approximately 0.3% prevalent in the general population; in the elderly people over 60 years, it is ~1%, and ~3% in elders over 80 years [2]. The PD incidence rate is 8–18 person per 100,000 people per year [2]. The median age at onset is 60 years, and the mean duration of the disease progression, from diagnosis to death, is around 15 years [2]. This disease and its incidence rate are 1.5–2 times prevalent in men than in women [2]. Moreover, PD costs 2500 USD each year for medical treatments and up to 100,000 USD per patient for therapeutic surgery [3]. ALS, the third most common NDD and most common motor neuron disease, has an incidence about rate of 1.9 people per 100,000 people per year [4,5]. In America, 30,000 people suffer from ALS, 30,000 from HD and o million from PD [6]. Because NDDs develop primarily in mid-to-late life, the incidence rate is expected to rise with the increasing aging population. By 2030, one of five Americans are expected to be over the age of 65, and over 12 million Americans may suffer from NDDs 30 years from 2020 [7]. Thus, early screening and treatments for NDDs should be achieved to meet the growing demand on preventive medicine. NDDs can influence many kinds of body activities, such as heart regulation, respiration, speech, mental functioning, balance and movement. Because general motions such as flexion and extension of the two lower limbs are controlled by the central nervous system, especially basal ganglia, the gait of the patient with an NDD may become abnormal (different gait pattern from the healthy subject) owing to a malfunctioning motor neuron [8]. ALS, also called motor neuron diseases, causes the death of neurons that control voluntary muscles; this condition results in stiff muscles, muscles twitching and gradually worsening weakness attributable to muscles decreasing in size [9,10,11]. HD is a hereditary disorder that results in the death of brain cells; thus, lack of coordination, an unsteady gait and uncoordinated and jerky body movements will become more apparent [12,13,14]. PD is a long-term degenerative disorder of the central nervous system; it mainly affects the motor system, and its early symptoms include shaking, rigidity, slow movement and difficulty of walking [15,16,17]. Thus, the gait is affected by NDDs. As a result, information about gait is used to analyze movement in HC (HC) subjects and other subjects with different kinds of diseases. The gait analysis is very useful in understanding movement disorders caused by NDDs and it can be potentially used in presenting the non-invasive automatic classification method for NDDs.

Gait analysis is used to assess and treat individuals with conditions that affect their ability to walk, such as health, age, size, weight and speed. In previous studies, as shown in Table 1, research on gait analysis has been developed using the series of stride, stance or swing intervals, ground reaction force (GRF) and foot force. Wei Zeng and Cong Wang presented the gait dynamics method to classify NDDs via the deterministic learning theory [18]. Using statistical features and different classification models, Xia et al. proposed a classification method for gait rhythm between patients with neurodegenerative diseases and control subjects [19]. $\mathrm{Ertu}\overset{˘}{g}\mathrm{rul}$ et al. developed shifted one-dimensional local binary patterns to detect PD based on a vertical GRF (vGRF) [20]. Wu et al. measured signal fluctuations in the gait rhythm time series of patients with PD using entropy parameters to compute the approximate entropy (ApEn), normalized symbolic entropy and signal turns count parameter for stride fluctuations measurement [21]. Generalized linear regression analysis and support vector machine (SVM) were applied to perform nonlinear gait pattern classifications. Zhao et al. implemented dual-channel long short-term memory (LSTM)-based multi-feature extraction on gait for diagnosis of NDDs [22]. They designed a dual-channel LSTM model to merge time series and force series recorded from NDDs patients for whole gait understanding. Suleyman Bilgin researched about the impact of feature extraction to classify ALS patients among those with NDDs and the HC subjects [23]. Compound force signal, the input signal, was utilized for feature extraction using a 6-level discrete wavelet transform with different types of wavelet techniques. The obtained features were validated using 20 trials for 5-fold cross-validation in linear discriminant analysis (LDA) and Naïve Bayesian classifier (NBC). Pham (2017) proposed a novel method for gait analysis by transforming time series data sequence into images from which texture analysis methods and texture features of a gait can be extracted [24]. In addition, the existing literature only focused on two-classification (e.g., HC vs. ALS, HC vs. HD, HC vs. PD), multiclass classification (e.g., HC vs. ALS vs. HD vs. PD) at the same time had never been studies before. This study not only focused on two-class gait classification, but also focused on multiclass (four-class) gait pattern classification.

Less involvement of the raw physiological signal analysis during gait analysis and adoption of state-of-the-art deep learning classifiers can be noted in the existing studies. In some literature reports (presented in Table 1), NDDs gait classifications were developed using features such as series of stride/stance/swing intervals, which are the processed features of the raw physiological signal [18,19,21]. To investigate new differences among HC and NDDs (ALS, HD and PD) was the first research aim of this study. Deep learning classifiers, which can automatically construct representations of the data, was an appropriate technology to investigate the difference in the raw physiological signal between HC and NDDs (ALS, HD and PD). To transform the raw physiological signal into image-like features and then to utilize deep learning classifiers to develop the NDDs classification algorithm was the second research aim of this study. Thus, the recurrence plot was used to transform the vGRF into a recurrence plot image. Convolutional neural network (CNN), a famous deep learning classifier, was used to extract the features from the recurrence plots and classify the features of HC and NDDs (ALS, HD and PD) gaits. The utilization of existing methods combination in the proposed method, such as recurrence plot and CNN, is for transforming the raw physiological signal from 1-dimensional space (time domain) to 2-dimensional space (spectrogram, time–frequency domain) and performing the feature extraction in order to bring out the most important pattern visualization.

Although some existing methods were employed in this research, the proposed method came out with novel concept in gait analysis for NDDs identification based on extracted pattern visualization of a raw physiological signal. The novelty of this research was to develop a sophisticated approach for the NDDs classification using recurrence plot’s pattern visualization and a deep learning algorithm instead of the statistical features [19,20,21] and machine learning algorithm [19,20,21,23,24] as previously stated. The extracted pattern features were automatically generated using the deep learning algorithm. Contrasting with the extracted statistical features, over the pattern visualization of the recurrence plot, the gait abnormalities within the NDDs patients can be directly and effectively identified and distinguished from the gaits of the HC. The main objective of this study is to develop a classification method to help physicians in screening NDDs patients based on the vGRF data. In particular, this method will help determine if any of the three types of NDDs (ALS, HD and PD) can interfere with the patient’s ability to manage the propulsion of two feet. The method will also help determine if the significant differences in vGRF denote specific diseases the patient suffers. The right foot (RF), left foot (LF) and compound foot (CF) is obtained from the summation of RF and LF) force data of NDDs and HC subjects are used as the input to the algorithm. Then, feature transformation using a recurrence plot is applied to the input to create new features (gray-level texture image of recurrence plot) using the existing ones. For classification improvement, the principal component analysis (PCA) was applied to the gray-level texture image of recurrence plot by choosing the principal components (PCs) of the features. The PCs of HC and NDDs subjects are divided into training and testing sets. The estimators were built by training the training sets and by comparing the estimators with a test set of HC or NDDs to be classified; some parameters of classification were generated. CNN has successfully been applied in this study to extract the features and classify HC and NDDs in the classification stage (training and testing phase). The proposed method can effectively classify gait patterns among HC, ALS, HD and PD groups in neurodegenerative diseases.

## 2. Materials and Methods

In the proposed algorithm, the vGRF data of NDDs from the database of PhysioNet was utilized [25]. The raw data were obtained using force-sensitive resistors, with the output roughly proportional to the force under the foot [26]. The RF/LF/CF force data of HC, ALS, PD and HD was used as the input of the algorithm. In the proposed algorithm, the first step was to perform data preprocessing, remove corrupt data, and separate the original 5-min data into 10 s of consecutive data. Thereafter, the feature transformation method was applied to transform the signal into the image-like recurrence plot to emphasize and visualize the existing features. PCA was used to select and enhance the important features to improve the classification result. The deep-learning-based algorithm was applied to classify NDDs. A CNN was chosen as the classification model because it is efficient and robust in image classification. Lastly, a cross-validation method was employed to validate the trained model, leave-one-out cross-validation (LOOCV). Figure 1 shows the flowchart of the proposed method.

### 2.1. Neurodegenerative Diseases Gait Dynamics Database

The vGRF database used in this research (PhysioNet Gait Dynamics in Neurodegenerative disease) was made available online in the PhysioNet database by Hausdorff et al. [25]. The database comprised 64 recordings of information from 13 patients with ALS, 15 with PD, 20 with HD and from 16 HC subjects. There were two types of data recorded in this database: 1) raw data of gait vGRF series and 2) gait cycle patterns derived from the vGRF. The vGRF signal comprised LF force and RF force data. Within the gait cycle patterns, the contents were left stride interval(s), right stride interval(s), left swing interval(s), right swing interval(s), left swing interval (percent of stride), right swing interval (percent of stride), left stance interval(s), right stance interval(s), left stance interval (percent of stride), right stance interval (% of stride), double support interval(s) and double support interval (percent of stride). Only the vGRF signal data were used in the analysis since the main research purpose is to investigate new features from raw data. The classic processed gait patterns (such as left stride interval) are not considered.

Table 2 shows the demographics of the database subjects, including gender, age, height, weight, gait speed, and a measure of disease duration for ALS or severity for PD and HD. For the HC subjects, an indicator of 0 is used. For the ALS patients, the value describes the duration in months since the disease diagnosis. For the PD patients, the Hoehn and Yahr scale stages 1 through 5 is used [27], where a higher scale represents more severe disease. For the HD patients, the total functional capacity measure is applied, where a lower score exhibits more advanced functional impairment.

The vGRF signal data in this database was obtained using force-sensitive resistors in the insole, with the output proportional to the force under the foot. The transducer of the insole was a conductive polymer layer sensor with altered resistance when loaded. The sensor was selected based on various reasons: thickness of <0.05 in, temperature insensitivity, a fast-dynamic response, the ability to restrain an overload and an electronically easy interface. Two 1.5 in^2^ force-sensitive resistors were used and the sensors were taped to an insole used to place them inside the shoe. The insole was made from the manila folder by tracing an outline of the foot onto it and then cutting out the tracing. One sensor was located to the anterior portion of the insole, under the toes and the metatarsals, and the other sensor was at the opposite end, under the heel. The two footswitches were connected in parallel and served as one large sensor (the outputs from these two footswitches were added up). Then, the analog signal was digitized and analyzed using software [26].

### 2.2. Data Preprocessing

#### Time-Windowing Process (10-s Window Length)

The original data were collected for 5-min per subject. To eliminate the impact of the initial walking period of each subject, the first 20-s length of data was removed. In the proposed algorithm, window function, as a mathematical term that is zero-valued outside some selected interval to separate the original data into several consecutive data sets, was used. In this step, the rectangular window function [28], as denoted in and overlapping two neighboring time windows for 6.66-s, as depicted in Figure 2,were applied to an original 5-min-long signal to obtain the 10-s-long successive signals. The overlapping time window method was successfully applied in several studies [29,30,31,32].

The total number of subject data was 64, with 16 data for HC, 13 data for ALS, 15 data for PD and 20 data for HD. However, after this time-windowing process was employed, 1312 data for HC, 1066 data for ALS, 1230 data for PD and 1640 data for HD were obtained, which meant that 5248 data were available for the training model, as shown in Table 3. The number of vGRF data samples (n) after data preprocessing can be calculated as:(1)$$n=\left(\frac{\ell -TW}{d}+1\right)\times T$$

ℓ is the data length, TW indicates the time window length (10-s), d is the distance between two windows data (2/3 of time-window size as the result of overlapping two neighboring time windows for 6.66-s) and T is each group total samples. Each of HC, ALS, PD and HD subject has 5-min (300-s) length of data and the first 20-s of data were removed. The remaining data length is 280-s for each subject (ℓ=280). The time window length is 10-s (TW=10) and the distance between two windows data is 3.33-s, since the overlap window for each signal is 6.66-s. There are $\frac{\left(280-10\right)}{3.33}+1=82$ data samples for each subject and finally, the total samples are 82×16=1312 data samples for HC (16 subjects), 82×13=1066 data samples for ALS (13 patients), 82×15=1230 data samples for PD (15 patients) and 82×20=1640 data samples for HD (20 patients).

There were several benefits regarding the use of the time-windowing process. First, it was useful to obtain more data for the deep learning model to obtain an accurate prediction. Second, use of the time-windowing process meant shorter signal data were gained. In the real-life situation, this is related to the patient’s convenience while performing data collection. If a 5-min length of data is used to obtain sufficient data, the patients need to walk for at least 5 min, which can be time-consuming and inconvenient for NDDs patients. The potential for the patient to be injured, because of the fall risk factor, is increased if the data collection time is longer. In case of the 10-s-long data, patients will only need to walk for 10 s and will also get the reliable prediction of the disease faster (rapid NDDs detection algorithm). Third, in shorter vGRF signals, more detailed texture and pattern visualization of gait abnormalities can be observed.

### 2.3. Recurrence Plot

A recurrence plot was utilized as the feature transformation method. The recurrence plot is a good visualization tool for capturing hidden dynamics of nonlinear time series. The original signal was transformed into a two-dimensional image by the recurrence plot [33]. The useful and important information from a complex signal or complex system can be displayed, and the texture patterns can also be further analyzed. For the recurrence plot used in the proposed algorithm, the principle is explained as follows: Let $X=\left\{\;x_{1\;},x_{2},\;\ldots ,\;x_{n-1\;},\;x_{n\;}\right\}$ be a set of force series in a record of gait force signal, where n denotes the data point. A recurrence plot can be constructed as follows:(2)$$P\left(i,j\right)=\frac{\left|x_{i}-x_{j}\right|}{max\left(X\right)}$$
where $P\left(i,j\right)$ is a pixel with the coordinates $\left(i,j\right)$ in a recurrence plot, for i=1, 2, …, n, and j=1, 2, …, n. $P\left(i,j\right)$ is similar or close to a state pair ($x_{i},\;x_{j}$) in meaning. The data were normalized by dividing with the maximum value of X so that all the pixel values will be ranged from 0 and 1. If the value of $x_{i}\;$and $x_{j}$ are more similar, the pixel shown on the recurrence plot will be closer to black; in contrast, if the value of $x_{i}\;$and $x_{j}$ differ, the pixel shown on the recurrence plot will be closer to white. With the recurrence plot, a gray-level image can be obtained, which represents the complexity and regularity of the input signal by the rendered colors and the texture pattern. The not-so-obvious periodic features of the original signal can also be emphasized and visualized.

The data of each subject in each group that had already been processed by the time-windowing process as the input signal to construct a recurrence plot. As shown in Figure 3, the different texture patterns can be observed through the recurrence plots of different types of groups (HC, ALS, PD and HD), which indicate that the generated images were suitable to be classified by deep learning algorithms.

### 2.4. Principal Component Analysis

The main idea of a PCA is to perform dimensionality reduction of a dataset containing a major number of interrelated variables while resisting as much as possible of the variation present in the dataset [34]. This is acquired by transforming the dataset into a new set of variables, the principal components (PCs), which decorrelates the variables that are ordered.

The PCA method in this research is defined mathematically using the following steps (described as a flowchart in Figure 4): Consider that a matrix, $X={\left[P_{1};P_{2};P_{3};\ldots ;P_{i}\right]}^{T}$, is constructed by the gray-level texture images of all NNDs and HC, where P is a row vector consisting of the pixels of a gray-level texture image of NDDs or HC and i is the number of gray-level texture images of all NDDs and HC. The PC is built using the equation:(3)$$C=X^{T}X$$
It is also called a covariance matrix of the matrix X to subsequently find its eigenvalues and eigenvectors. Then, the W matrix, an m×m matrix of weights whose columns are the eigenvectors of C, is obtained. Finally, the matrix of extracted feature F can be described as the full PCs’ decomposition of X and can, therefore, be shown as:(4)F=XW
Because PCA was applied as the feature enhancement and the input was an image, the full PCs of each sample was selected to maintain the important texture and pattern features for visualization. The purpose of using PCA as the feature enhancement in this proposed method is to enhance the between-class separability and minimize the within-class separability of datasets. It was intended to improve the classifier performance in classifying the data points into the correct group.

### 2.5. Convolutional Neural Network

A CNN is composed of one or more convolutional layers (often with subsampling and pooling layers), which is then followed by one or more fully connected layers as in a basic multilayer neural network (deep learning) [35]. The architecture of a CNN is built to benefit from the 2D structure of the input (image or signal). This is accomplished by local connections and involves weights followed by any pooling function that results in translation-invariant features. Another advantage of CNN is that it is simpler to train and has significantly fewer parameters than other fully connected networks with the same number of hidden layers. The main reason for using a CNN in the proposed method is to distinguish the difference between the gray-level texture image representation of vGRF from HC and NDDs (ALS, HD and PD) subjects. The concept of using recurrence plot and CNN to extract and classify NDDs vGRF data are never found in our literatures. A pre-trained AlexNet [36] was used in this study in order to meet a balance point between classification accuracy performance (significant improved compare to the classical CNN such as LeNet [37]) and computation time (much less time consumption compared to the state-of-the art CNN such as GoogLeNet [38] or ResNet [39]).

A pretrained AlexNet CNN was utilized from MATLAB R2018a Deep Learning Toolbox^TM^ in the system [36]. Kirzhevsky et al. trained a large and deep convolutional neural network, called pretrained AlexNet, with 1.2 million high-resolution images into 1000 different labels on multiple GPUs. The error rate was approximately 1.7%. As the result, the pretrained AlexNet has learned rich feature representations for a wide range of images as the input. The architecture comprises 25 layers, including an input layer, five convolution 2D layers, seven ReLU (activation function) layers, two cross-channel normalization layers, three max-pooling 2D layers, three fully connected layers, two dropout layers (for regularization), a softmax layer (normalized exponential function) and an output layer. The input of the pretrained AlexNet in the proposed method is the gray-level texture images of the vGRF data yielded by the recurrence plot. There are two methods in fine-tuning a pretrained AlexNet: transfer learning and feature extraction. The feature extraction method was selected because it is an easy way to apply the pretrained networks without spending much time (i.e., faster than the transfer learning method) and many attempts for training. This method only applies to earlier fully connected layers and uses an SVM for classification. Earlier layers characteristically extract fewer, shallower features, have higher spatial resolution and a larger total number of activations. On the contrary, deeper layers contain higher-level features, constructed by the lower-level features of earlier layers. The proposed feature extraction method only utilized 20 layers out of 25 layers’ pretrained AlexNet CNN, from input layer (total input = 5248 images) to the fully connected layer ‘fc7′, in order to get the higher-level features (depicted in Figure 5).

The convolutional layer plays the most important role in how CNNs work. This layer is composed of a set of kernels (learnable filters) as parameters, which contain a small receptive field, but are prolonged through the full depth of the input. When the data pass through the convolutional layer, each kernel is convolved across the spatial dimensionality of the input (width and height of the input volume), calculating the dot product and producing a 2D activation map. The filters in the convolutional layers are edge detectors and color filters. The ReLu (rectified linear unit) layer utilizes the non-saturating activation function $f\left(x\right)=\max\left(0,x\right)$, such as sigmoid $\sigma \left(x\right)={\left(1+e^{-x}\right)}^{-1}$, to the output of the activation generated by the previous layer. Another vital concept in CNNs is pooling, which is usually referred to as nonlinear downsampling. The aim of the pooling layer is to perform a dimensionality reduction and to minimize the number of parameters and the complexity of model computation. This layer takes action in the input of each activation map and scales the input dimension using the “MAX” function, hereafter called the max-pooling layer. Eventually, after some convolutional and max-pooling layers, the fully connected layers will attempt to generate class scores from the previous activations to be used for classification; the same roles that they play in traditional forms of artificial neural networks. Neurons in this layer have connections to all activations from the previous layer.

### 2.6. Cross-Validation

Cross-validation is a statistical method used to assess and compare learning algorithms by dividing data into two groups: one used to learn or train a model (training set) and the other used to validate the model (testing or validation set) [40]. The training and testing sets must cross over in consecutive rounds such that each data point has an opportunity to be validated. There are two main purposes for applying cross-validation: First, the performance of the learned model from available data using one algorithm can be investigated. In other words, it is used to quantify the generalizability of an algorithm. The second purpose is to evaluate the performance of two or more different algorithms to discover the best algorithm for the available data or, alternatively, to compare the performance of two or more variants of the parameterized model. Leave-one-out cross-validation (LOOCV) is a special case of k-fold cross-validation, where k equals the number of data points. In other words, in each iteration, almost all the data points, except for a testing data point, are used for learning (training), and the model is validated on that single data point. An accuracy estimation obtained using LOOCV is known to be almost unbiased, but it has high variance, inferring unreliable estimates.

## 3. Experimental Results

The experiments were executed using MATLAB R2018a software on an NVIDIA GeForce GTX 1060 6 GB computer with 24 GB RAM. The experiment results consist of included two-class and multiclass classifications. Two-class classification results are classification of the NDD and the HC groups and classification among the NDDs. The classification accuracy for (HC vs. ALS), (HC vs. HD), (HC vs. PD), (ALS vs. PD), (ALS vs. HD), (PD vs. HD) and (NDDs vs. HC) The multiclass classification, a four-class gait classification among HC, ALS, PD and HD was conducted. The classification algorithm in this study comprises three processes: a preprocessing, feature transformation and classification. In the preprocessing, two different window length: 10-s and 5-min were selected as the time window of the gait signal for classification. The objective of selecting a 10-s window is to develop an NDDs gait classification with short observation signal length; the objective of selecting a 5-min window is to compare the performance to that of the existing literature. In the feature transformation process, the time-domain vGRF data are modified into an image using a recurrence plot. The principal component analysis (PCA) is used in this stage for feature enhancement. Lastly, the convolutional neural network (CNN), as a deep learning classifier, is employed in the classification process and evaluated using the leave-one-out cross-validation (LOOCV). The average execution time of this study is shown in Table 4. The accuracy (acc.), sensitivity (sens.), specificity (spec.) and an AUC value of the proposed method were measured as the parameters for evaluation. The definition of the evaluation parameters is provided in [41].

When selecting between two or more diagnostic tests, Youden’s index is generally applied to evaluate the effectiveness of an overall diagnostic test [42]. Youden’s index is a function of sensitivity and specificity, where the index ranges between 0 and 1, with a value close to 1 means that the diagnostic test effectiveness is relatively high and the test is perfect, and a value close to 0 represents limited effectiveness, where the test is useless. The Youden’s index (*J*) is described as the sum of the two fractions indicating the measurements correctly diagnosed for the diseased group (sensitivity) and HC (specificity).
(5)$$J=\left(sensitivity+specificity\right)-1$$

This index was employed to select better classification results among the LF, RF and CF of vGRF data. The classification results are given in two parts: 1) two-class classification and multiclass classification. In two-class classification, the results include classification of the NDD and HC group, classification among the NDDs and classification among the NDDs. The multiclass classification includes the classification among the HC, ALS, PD and HD at the same time. Even though the application of the CNN in the multiclass classification has been used in different domains of medical research such as in [43,44,45,46,47], the multiclass classification is the novelty of this study and existing literature did not do the multiclass classification.

### 3.1. Two-Class Classification

The two-class classification in this study includes three sub-study. There are (1) classification of the NDD and healthy controls group, (2) classification among the NDDs, (3) classification of All NDDs in one group and healthy controls Group. The purpose of studying classification of the NDD and healthy controls group is to examine the performance of the proposed algorithm to certain NDD (ALS, HD, PD). The objective of classification among the NDDs is to check how the proposed method perform with various NDDs classification. Finally, all NDDs data were combined into one class and discriminate with that of the HC class.

#### 3.1.1. Classification of the NDD and Healthy Controls Group

In this classification situation, there were three kinds of different classification tasks, such as ALS vs. HC, HD vs. HC and PD vs. HC. There were 12 ALS, 20 HD and 14 PD patients as well as 16 HC subjects who were observed in all classification situations, but the input signal for the proposed method was dependent on the window size in the time-windowing process. For the 10-s time-window size, there were 1312 data windows of HC, 984 data windows of ALS, 1148 data windows of PD and 1640 data windows of HD, which was a total of 5084 data. For the purpose of comparison, the 5-min time-window size was also employed. The detailed classification results are given in Table 5 and Table 6.

#### 3.1.2. Classification among the NDDs

In this study, a concept for classification among the NDDs was developed, for example, ALS vs. HD, PD vs. ALS and HD vs. PD. The main purpose of this classification was to provide intra-class separation efficiency (the NDD group: ALS, HD and PD), i.e., whether ALS, HD and PD could be easily separated or not. It was concluded that the ALS group could be readily distinguished from the HD and PD groups. HD and PD were not easy to separate. The HD vs. PD classification performance, accuracy, sensitivity, specificity and AUC value, were less compared to ALS vs. HD and PD vs. ALS in 5-min time-window size. This occurred because HD and PD disorders are caused by the degeneration of basal ganglia, and the gait abnormality symptoms of HD and PD patients are almost identical [48]. However, this issue can be surmounted by the proposed algorithm using 10-s time-window size. The complete classification results of this classification situation are shown in Table 5 and Table 6. Table 5 and Table 6 can reveal that the proposed algorithm can perform good in the classification between HC and any one of NDDs or classification between NDDs.

#### 3.1.3. Classification of All NDDs in One Group and Healthy Controls Group

In NDDs vs. HC classification, the ALS, HD and PD patients’ vGRF datasets were merged in one group, for which the total number of NDD datasets was dependent on the time-windowing size. The experimental results for this classification situation are shown in Table 5 and Table 6.

### 3.2. MultiClass Classification

The multiclass classification is closer to the clinical application, since the physician will not have preliminary information about whether the patient is suffering from ALS, HD or PD. The whole vGRF dataset was divided into four classes, based on the patients with the diseases (ALS, HD and PD) and healthy subjects. LOOCV was also applied in the multiclass classification for evaluation and validation approaches. The detailed classification results are presented in Table 7 and Figure 6.

## 4. Discussion

The gait analysis of different subjects is discussed in this section based on the texture analysis of the recurrence plot. In the original vGRF signal data, it was not easy to observe some key features by naked eye. However, after the signal was transformed into the recurrence plot, different periodic features in the original signal could be extracted and shown as different texture patterns on the plot. The different texture patterns between each kind of NDD and HC subjects can be easily pointed out. With the method of recurrence plotting, the special features of the original vGRF signal in each type of subject can be emphasized and visualized into a plot. It brings the benefit of the follow-up deep learning algorithm CNN, which is outstanding for image recognition.

### 4.1. Healthy Control

The texture patterns in the recurrence plots of HC subjects depicted in Figure 3a are orderly and regular. As shown in figure, there are roughly two different kinds of black squares in the plot, one is a bigger black square and the other is the smaller black square. The larger black squares represent the stance phase during the gait cycle, and the smaller black squares represent the swing phase. Both kinds of squares appear repeatedly and regularly in the plot, and the side length is almost the same in the whole the plot, whether the squares are bigger or smaller. This means that each stance interval and each swing interval of the original signal are consistent and regular, which corresponds with the characteristics of healthy people.

### 4.2. Amyotrophic Lateral Sclerosis

The texture patterns are much more complicated in the recurrence plots of ALS subjects, as shown in Figure 3b. There are also two kinds of black squares, a bigger one and a smaller one, which represent the stance phase and the swing phase, respectively. However, in the plots of ALS subjects, the size of bigger black squares is obviously bigger than those of HC subjects, which means that the stance intervals of ALS subjects are longer than the stance interval of HC subjects in comparison. Furthermore, there are clear “cross-like” patterns appearing in the bigger black squares, showing the feature of a double peak in the stance phase of the original signal.

### 4.3. Parkinson’s Disease

The recurrence plots of PD subjects, shown in Figure 3c, are similar to those of HC subjects. However, by careful observation, it can be seen that the black square in the plot of HD patients is irregular. There are no obvious two kinds of black squares that can be pointed out. This means that the stride intervals and the swing intervals of PD subjects are more irregular than the HC subjects.

### 4.4. Huntington’s Disease

The recurrence plots of HD subjects are the most arbitrary and irregular plots of the NDDs, indicated in Figure 3d. There are also two kinds of squares that can be observed. The first kind has a clear cross-like pattern within. It represents the stance phase with the feature of a clear double peak. The other kind, without the cross-like pattern, represents the swing phase. The size of these two kinds of squares is similar, which shows that there is no clear difference between the length of the stance interval and the length of the swing interval for the HD subjects.

### 4.5. Classification Performance Comparison to Other Literature Based on PhysioNet Gait Dynamics in Neurodegenerative Disease Database

In order to examine performance of the proposed algorithm, four literatures [18,19,22,24] are selected to compare the algorithm performance. [18,19,22,24] adopted PhysioNet Gait Dynamics in Neurodegenerative disease [26] which is the same as this study. [18] employed stance and swing intervals series of left and right foot to do the two-class classifications (including classification of ALS vs. HC, HD vs. HC and PD vs. HC). [22] utilized two kinds of data as input, gait pattern data (stance and swing intervals series of left and right foot) and gait force data (vGRFs). Two-class classification including HC vs. ALS, HC vs. HD, HC vs. PD and NDDs vs. HC were compared using LOOCV as the cross validation. [19] adopted five independent gait parameters including, stance interval of LF and RF, stride interval of LF and RF and double support interval, were selected. Two-class classification comparison of ALS vs. HC, PD vs. HC, HD vs. HC and NDD vs. HC using LOOCV were presented. In [24], only RF gait force signal was chosen as the input of the algorithm. [24] presented sensitivity, specificity, AUC value, and the accuracy of HC vs. HD, HC vs. PD and HC vs. ALS classifications using LOOCV as the evaluation method. The comparison of [18,19,22,24] to the proposed algorithm is given in Figure 7, Figure 8, Figure 9 and Figure 10. The proposed method obtained a satisfactory performance of NDDs classification compared with [18,19,22,24]. In summary, the proposed method outperforms [18,19,22,24] in HC vs. ALS, HC vs. PD and NDDs vs. HC classification. In HC vs. HD classification, the proposed method cannot achieve the accuracy as high as that of the [19,24] (98.85% vs. 100%). However, for the length of the input data, the proposed method only used 10-s length data with high classification performance, it indicates that the proposed method can be categorized as an effective and rapid NDDs screening algorithm. It is also more appropriate for patient data collecting, the patients do not need to walk in a certain long period of time (5-min) so the fall incidence can be minimized.

### 4.6. Limitations of the Proposed Method

Even though the proposed method obtained importance-performance evidence, there are some limitations that has to be improved. Limited number of data were used as the input of the proposed method and it was from the existing online database, to collect clinical data from HC and NDDs subjects is the future work of this study. The other major drawback is the deployment of the patients’ age and disease severity level were not investigated well. These factors will influence the emergence of different gait abnormalities that affect the gait pattern visualization and the classification performance of deep learning algorithm. Based on these limitations, there are several major directions for improvement that could be carried out. First, since the performance obtained using the proposed method used an existing database, clinical data should also be obtained for the purpose of verification and to resolve the limitations of the current database (the limited number of NDD patients). Second, long-term data collection for monitoring NDDs progression is meaningful for the treatment of the NDD patient since the gait pattern of NDD patients should be changed in the long-term disease progression. Third, in order to assure the clinical meaning, the NDD gait phenomenon based on a gray-level texture image should be discussed with physicians. Fourth, other input data (such as kinetic data, temporal data, step length and cadence) and classifiers should be applied in order to confirm and compare the effectiveness of pattern visualization and recognition based on the use of a gray-level texture recurrence plot image in NDD detection applications.

## 5. Conclusions

A novel deep-learning-based NDD detection algorithm using a recurrence plot based on vGRF signal data were developed. Pattern visualization and recognition of the recurrence plot image-like made it possible to successfully differentiate between the gait phenomenon of NDD patients and HC. After the original signal was transformed, feature enhancement using PCA was applied to increase the between-class separability and reduce the within-class separability. In order to evaluate the CNN classification process, LOOCV was performed, and four parameters were generated, including accuracy, sensitivity, specificity and the AUC value. The classification accuracy for (HC vs. ALS), (HC vs. HD), (HC vs. PD), (ALS vs. PD), (ALS vs. HD), (PD vs. HD) and (NDDs vs. HC) were 100%, 98.41%, 100%, 95.95%, 100%, 97.25% and 98.91%, respectively. In the multiclass classification, a four-class gait classification among HC, ALS, PD and HD was conducted and the classification accuracy of HC, ALS, PD and HD were 98.99%, 98.32%, 97.41% and 96.74%, respectively. The proposed method can achieve high accuracy compare to the existing results, but with shorter length of input signal (Input of existing literature using the same database is 5-min gait signal, but the proposed method only needs 10-s gait signal). As a result, the proposed method was able to achieve the highest performance for more than 98.41% of the parameters being evaluated and achieved superior performance in comparison to NDD detection state-of-the-art methods found in the literature.

## Figures and Tables

**Figure 1 sensors-20-03857-f001:**
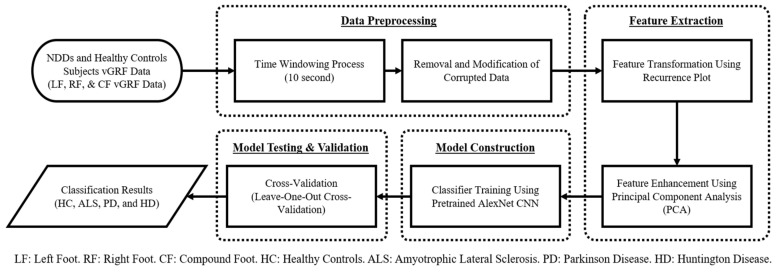
Flowchart of the proposed NDD detection algorithm using recurrence plot as the feature transformation.

**Figure 2 sensors-20-03857-f002:**
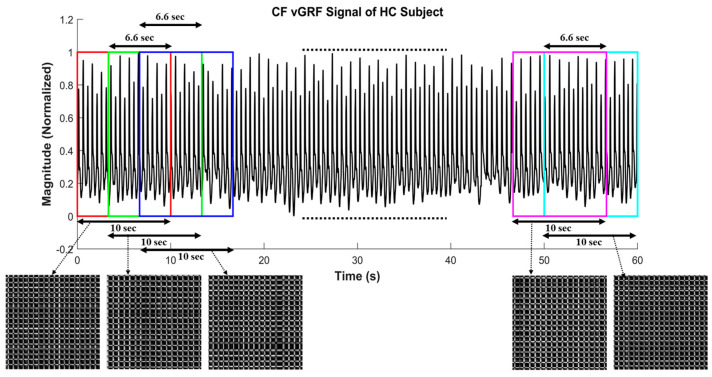
Illustration of overlapping 10-s time-window size for 6.66 s (2⁄3 of time-window size), including the corresponding recurrence plot of each 10-s window length compound foot (CF) vertical ground reaction force (vGRF) signal of healthy controls (HC) subject.

**Figure 3 sensors-20-03857-f003:**
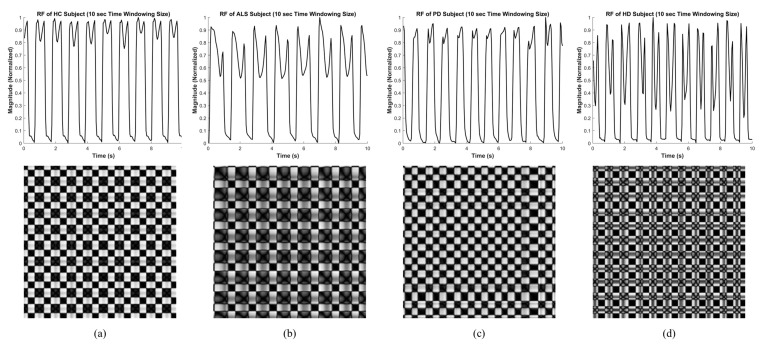
Image feature extracted using the recurrence plot of the right foot vGRF signal of healthy control and neurodegenerative diseases subjects at 10-s time-windowing size (image resolution: 227×227). (**a**) Healthy subject (HC); (**b**) amyotrophic lateral sclerosis patient (ALS); (**c**) Parkinson’s disease patient (PD); (**d**) Huntington’s disease patient (HD).

**Figure 4 sensors-20-03857-f004:**
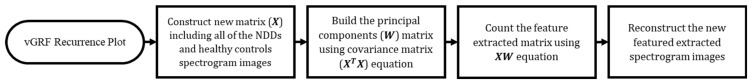
Flowchart of new feature extracted reconstruction using principal component analysis (PCA) as feature enhancement purpose.

**Figure 5 sensors-20-03857-f005:**
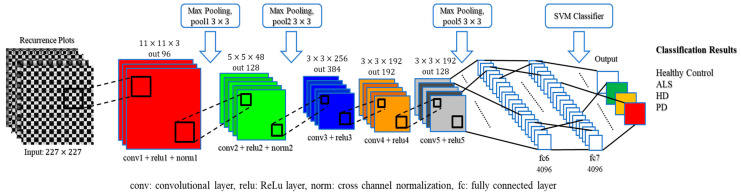
Architecture of the proposed method convolutional neural network (CNN).

**Figure 6 sensors-20-03857-f006:**
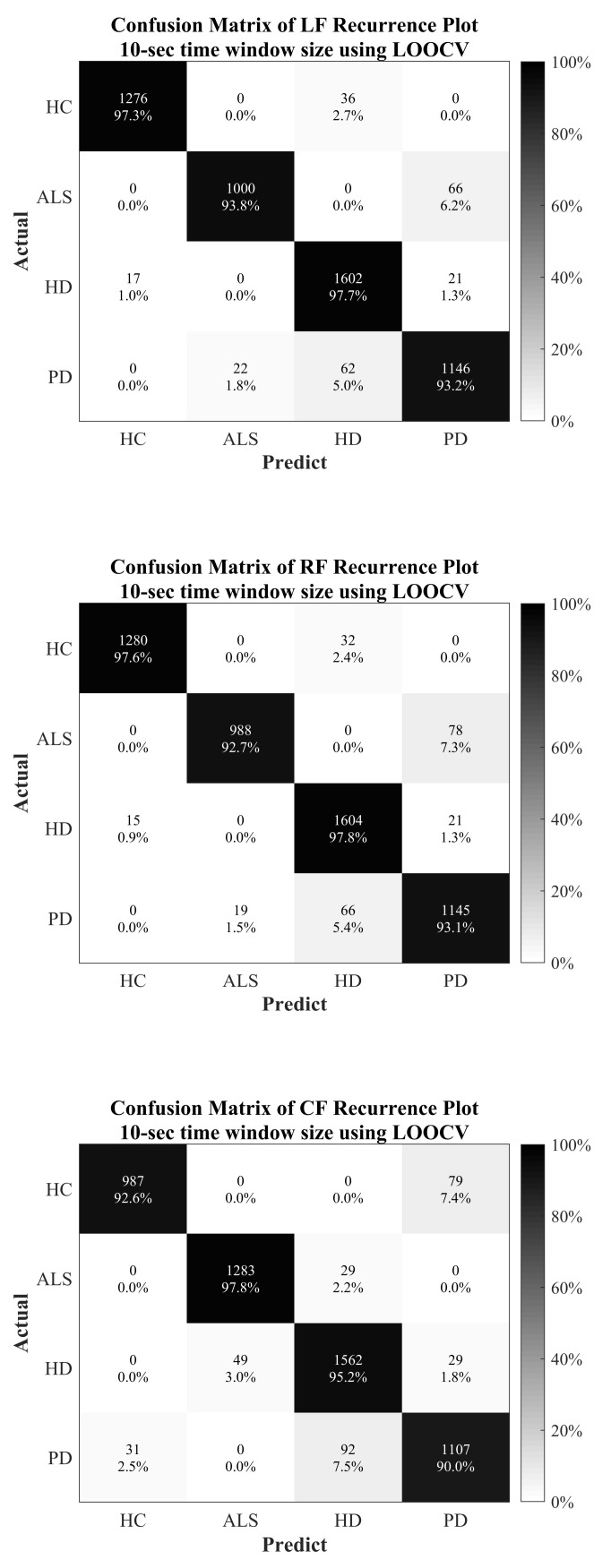
Confusion matrix for multiclass classification using LOOCV.

**Figure 7 sensors-20-03857-f007:**
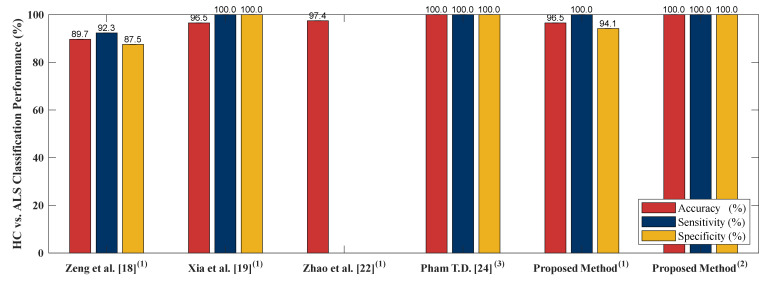
HC vs. ALS comparison result between the proposed method and existing literature. (**1**) 5-min data length; (**2**) 10-s data length; (**3**) 5-min data length; least squares support vector machine (LS-SVM).

**Figure 8 sensors-20-03857-f008:**
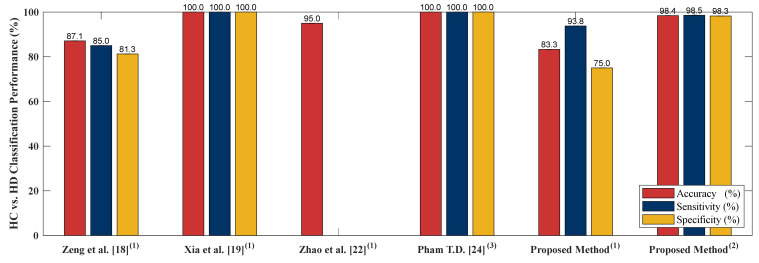
HC vs. HD comparison result between the proposed method and existing literature. (**1**) 5-min data length; (**2**) 10-s data length; (**3**) 5-min data length, LS-SVM.

**Figure 9 sensors-20-03857-f009:**
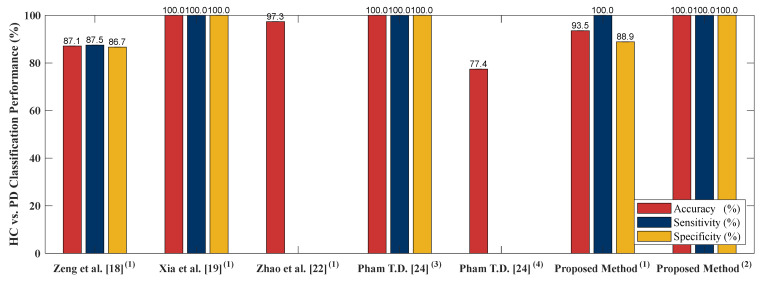
HC vs. PD comparison result between the proposed method and existing literature. (**1**) 5-min data length; (**2**) 10-s data length; (**3**) 5-min data length, LS-SVM, (**4**) 5-min data length, linear discriminant analysis (LDA).

**Figure 10 sensors-20-03857-f010:**
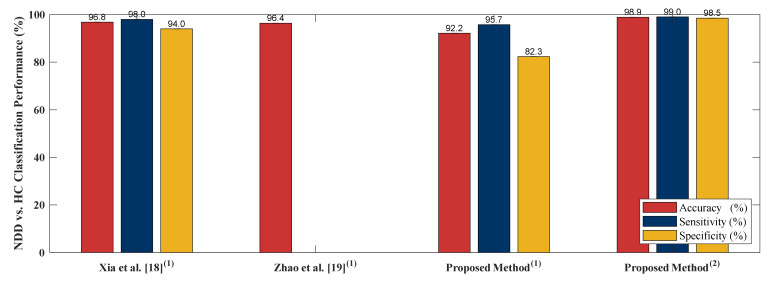
HC vs. NDD comparison result between the proposed method and existing literature. (**1**) 5-min data length, (**2**) 10-s data length.

**Table 1 sensors-20-03857-t001:** Summary of the neurodegenerative diseases (NDDs) gait classification literatures.

Literature	Summary of the Classification Algorithm		
	Feature Extraction	Classifier	Cross-Validation
[18]	Radial basis function (RBF) neural networks	RBF neural networks	All training all testing and LOOCV
[19]	Mean, standard deviation, max, min, skewness, kurtosis, Lempel-Ziv complexity, fuzzy entropy and Teager–Kaiser energy feature	Support vector machine (SVM), random forest (RandF), multilayer perceptron (MLP) and *k*-nearest neighbor (KNN)	LOOCV
[20]	shifted 1D-LBP	Bayes Network (BayesNT), naïve Bayes (NB), logistic regression (LR), MLP, Partial C4.5 decision tree (PART), RandF and functional tree (FT)	10-fold cross-validation
[21]	Approximate entropy (ApEn), normalized symbolic entropy (NSE), signal turns count (STC)	Generalized linear regression analysis (GLRA) and SVM	LOOCV
[22]	Dual channel LSTM	Dual channel LSTM	LOOCV
[23]	Discrete wavelet transform (DWT)	Linear discriminant analysis (LDA) and NBC	All training all testing and LOOCV
[24]	Fuzzy recurrence plot (FRP) + Gray-level co-occurrence matrix (GLCM)	Least squares support vector machine (LS-SVM) and LDA	LOOCV

**Table 2 sensors-20-03857-t002:** Demographics of the subjects in PhysioNet Gait Dynamics in Neurodegenerative disease database [25].

Class	Gender	Ages (Year)	Height (m)	Weight (kg)	Gait Speed (m/s)	Severity/Duration
	Male/ Female	(<50)/(50–70)/(≥70)				
HC	2/14	11/4/1	1.83 ± 0.08	66.81 ± 11.08	1.35 ± 0.16	0
ALS	10/3	4/7/2	1.74 ± 0.10	77.11 ± 21.15	1.05 ± 0.22	18.31 ± 17.82 ^1^
PD	10/5	1/7/7	1.87 ± 0.15	75.07 ± 16.9	1.0 ± 0.2	3 ^2^
HD	6/14	13/5/2	1.84 ± 0.09	73.47 ± 16.23	1.15 ± 0.35	8 ^3^

^1^ Duration in months since the disease diagnosis. ^2^ Hoehn and Yahr scale stages (1–5) in median quartile. ^3^ Total functional capacity scale (0–13) in median quartile.

**Table 3 sensors-20-03857-t003:** Number of vGRF data before and after data preprocessing.

Class	Number of vGRF Data	
	Number of Subjects (Original)	Samples of Time-Windowing Process (10-s)
HC	16	1312
ALS	13	1066
PD	15	1230
HD	20	1640
Total	64	5248

**Table 4 sensors-20-03857-t004:** Average execution time of the proposed method.

Proposed Action Methods	Execution Time (s)	
	10-s Length (5248 Input Samples)	5-min Length (60 Input Samples)
Feature transformation using recurrence plot	51.676	1.381
Feature enhancement using PCA	550.350	1.945
AlexNet CNN model training and testing using LOOCV	38,198.402	23.702

**Table 5 sensors-20-03857-t005:** Summary results for all two-class classification using leave-one-out cross-validation (LOOCV) for 10-sec time-window size.

Classification Tasks	10-sTime Window Size														
	Acc. (%)			Sens. (%)			Spec. (%)			AUC			J (Youden’s Index)		
	LF	RF	CF	LF	RF	CF	LF	RF	CF	LF	RF	CF	LF	RF	CF
ALS vs. HC	100	100	100	100	100	100	100	100	100	1	1	1	**1**	**1**	**1**
HD vs. HC	98.41	98.04	97.56	98.54	97.59	98.51	98.25	98.60	96.41	0.9839	0.9810	0.9746	**0.9679**	0.9619	0.9492
PD vs. HC	100	100	100	100	100	100	100	100	100	1	1	1	**1**	**1**	**1**
ALS vs. HD	100	100	100	100	100	100	100	100	100	1	1	1	**1**	**1**	**1**
PD vs. ALS	95.64	95.95	94.21	94.07	94.59	92.95	97.63	97.65	95.78	0.9585	0.9612	0.9437	0.9170	**0.9224**	0.8873
HD vs. PD	97.11	97.25	94.98	96.81	96.54	93.54	97.51	98.24	97.14	0.9711	0.9739	0.9534	0.9432	**0.9478**	0.9068
NDD vs. HC	98.86	98.91	98.93	99.01	99.04	99.44	98.38	98.53	97.43	0.9870	0.9878	0.9844	0.9739	**0.9757**	0.9687

Note: ****bold and underlined**** were selected by Youden’s index criteria as the best classification result and model.

**Table 6 sensors-20-03857-t006:** Summary results for all two-class classification using LOOCV for 5-min time-window size.

Classification Tasks	5-min Time Window Size														
	Acc. (%)			Sens. (%)			Spec. (%)			AUC			J (Youden’s Index)		
	LF	RF	CF	LF	RF	CF	LF	RF	CF	LF	RF	CF	LF	RF	CF
ALS vs. HC	96.55	96.55	86.21	100	100	90.91	94.12	94.12	83.33	0.9706	0.9706	0.8712	**0.9412**	**0.9412**	0.7424
HD vs. HC	77.78	83.33	77.78	83.33	93.75	92.86	72.22	75	68.18	0.7778	0.8438	0.8052	0.5555	**0.6875**	0.6104
PD vs. HC	93.55	90.32	80.65	100	100	90.91	88.89	84.21	75	0.9444	0.9211	0.8295	**0.8889**	0.8421	0.6591
ALS vs. HD	87.88	90.91	81.82	100	100	100	83.33	86.96	76.92	0.9167	0.9348	0.8846	0.8333	**0.8696**	0.7692
PD vs. ALS	71.43	71.43	71.43	76.92	73.33	70.59	66.67	69.23	72.73	0.7179	0.7128	0.7166	**0.4359**	0.4256	0.4332
HD vs. PD	82.86	77.14	68.57	79.17	75	69.57	90.91	81.82	66.67	0.8504	0.7899	0.6812	**0.7008**	0.5682	0.3624
NDD vs. HC	89.06	92.19	85.94	97.67	95.74	93.33	71.43	82.35	68.42	0.8455	0.8905	0.8088	0.6910	**0.7809**	0.6175

Note: ****bold and underlined**** were selected by Youden’s index criteria as the best classification result and model.

**Table 7 sensors-20-03857-t007:** Summary results for multiclass classification using LOOCV.

Classification Tasks	LF + 10-s Time Window Size				RF + 10-s Time Window Size				CF + 10-s Time Window Size			
	Acc. (%)	Sens. (%)	Spec. (%)	AUC	Acc. (%)	Sens. (%)	Spec. (%)	AUC	Acc. (%)	Sens. (%)	Spec. (%)	AUC
HC	98.99	97.26	99.57	0.9841	99.10	97.56	99.62	0.9859	98.51	97.79	98.76	0.9827
ALS	98.32	93.81	99.47	0.9664	98.15	92.68	99.55	0.9611	97.90	92.59	99.26	0.9592
HD	97.41	97.68	97.28	0.9748	97.45	97.80	97.28	0.9754	96.21	95.24	96.65	0.9595
PD	96.74	93.17	97.83	0.9550	96.49	93.09	97.54	0.9531	95.60	90	97.31	0.9366

Note: ALS = amyotrophic lateral sclerosis, HC = healthy control, HD = Huntington’s disease and PD = Parkinson’s disease.

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
