# Peer review of "Evaluation of Vertical Ground Reaction Forces Pattern Visualization in Neurodegenerative Diseases Identification Using Deep Learning and Recurrence Plot Image Feature Extraction"

_sensors, 2020, doi:10.3390/s20143857_

Round 1

Reviewer 1 Report

This paper is interesting to help guide earlier diagnosis and better management. I have only minor remarks

Abstract

Authors should indicate the number of plots retained in each group as this number may or may not give quality to the use of their algorithms.

Introduction

I don't think this study is relevant to introduce because it looks at a completely different area, that of long-term auto-correlations.

« Wu et al. measured signal fluctuations in the gait rhythm 75 time series of patients with PD using entropy parameters to compute the approximate entropy (ApEn), normalized 76 symbolic entropy, and signal turns count parameter for stride fluctuations measurement [21].”

The introduction is well written and precise. It has a good thread. I would suggest that the authors place their objectives at the end of the section.

Materiel and method

The data (Table 2) on disease severity for Hoehn and Yahr scale stages and Total functional capacity scale should be expressed in median quartiles because they are ordinal scales.

force-sensitive resistors in the insole assessed pression. How we calculate the force from its?

How many axes do you retain with PCA? Do you take into account the percentage of variance explained?

Author Response

Dear reviewer 1, thanks for your professional review comments and suggestions. please check our "reply to reviewer's comments" in the attached file. Thanks.

Reviewer 2 Report

In their manuscript, the authors propose the use Deep Learning methods, in particular Convolutional Neural Networks, for the classification of Neurodegenerative Diseases. 

Several detailed aspects of the methodology are missing, and make it hard to replicate the study. The most important ones to me are the following:

  1. In the processing stage, recurrence plots are generated from the 10 second samples of vGRF singals. Even when the exdpression used for the definition of the intensity at each pixel of the image is included, no detail on the image resolution is given. 
  2. How much reduction do you achieve by using PCA on the generated recurrence plots? 
  3. Even when some details of the CNN used for the study are provided in Section 2.5, only the type and amount of layers is given, but no indication on the order in which these are organized is provided. Probably a figure that shows the pattern which the 25 layers of the network are organized would be advisable.
  4. Why 25 layers and why those 25 layers? how was the process you followed to determine the network? Also related to point 2, how many inputs does your network have?
  5. How was the training of the network performed?. We only know about the separation between training and test data (from the LOOCV method).

Although the results tables are important, I would complement them with some graphs that, specially when comparing the results from your approach to previous efforts (data from table 8).

Even when the manuscript is generally well written, I found one text error and some word missuses that should be corrected. Line 167 has a missing reference. Line 207... are more similar (instead of identical). Line 208 ... value of xi and xj differ (instead of are differ). Line 262... without spending a lot of... (instead of expending).

I think references should be provided for AlexNet, LeNet, GoogLeNet and ResNet (lines 251 to 254).

Although a multi-class classification of NDD might have not been reported yet, the use of CNNs as base for multi-label classification has been used in different domains. Even the AlexNet seems to have been developed originally for such problem, thus a small review of such efforts would also add to justify the use of CNNs to this particular domain problem.

Author Response

Dear reviewer 2, thanks for your professional review comments and suggestions. please check our "reply to reviewer's comments" in the attached file. Thanks.

Reviewer 3 Report

The paper is technically sound, however, the proposed method is based on existing techniques and published ideas (i.e. recurrence plot, APC, CNN) without theoretical justification for its performance. It is suggested to discuss more this point and highlight the novelty of this paper.

1) to discuss and justify theoretically the performance the proposed method, which based on a combination of existing methods and published ideas.
2) also, to highlight the novelty of the proposed method.

Author Response

Dear reviewer 3, thanks for your professional review comments and suggestions. please check our "reply to reviewer's comments" in the attached file. Thanks.

Round 2

Reviewer 2 Report

I consider all the issues I previously raised have been addressed.

There is a typo in line 14, a single closing parenthesis... "summation of RF and LF)"